# The Local Release of Teriparatide Incorporated in 45S5 Bioglass Promotes a Beneficial Effect on Osteogenic Cells and Bone Repair in Calvarial Defects in Ovariectomized Rats

**DOI:** 10.3390/jfb14020093

**Published:** 2023-02-09

**Authors:** Juliani Caroline Ribeiro de Araújo, Leonardo Alvares Sobral Silva, Vinicius Almeida de Barros Lima, Tiago Moreira Bastos Campos, Paulo Noronha Lisboa Filho, Roberta Okamoto, Luana Marotta Reis de Vasconcellos

**Affiliations:** 1Department of Bioscience and Buccal Diagnose, Institute of Science and Technology, Sao Paulo State University, UNESP, Sao Paulo 12245-700, SP, Brazil; 2Department of Prosthodontics and Periodontology, Bauru School of Dentistry, University of Sao Paulo, Bauru 17012-230, SP, Brazil; 3Department of Physics, São Paulo State University, UNESP, Bauru 17033-360, SP, Brazil; 4Department of Surgery and Integrated Clinic, Araçatuba Dental School, São Paulo State University, UNESP, Araçatuba 16066-840, SP, Brazil

**Keywords:** biocompatible materials, bone regeneration, cell differentiation, osteoporosis, teriparatide

## Abstract

With the increase in the population’s life expectancy, there has also been an increase in the rate of osteoporosis, which has expanded the search for strategies to regenerate bone tissue. The ultrasonic sonochemical technique was chosen for the functionalization of the 45S5 bioglass. The samples after the sonochemical process were divided into (a) functionalized bioglass (BG) and (b) functionalized bioglass with 10% teriparatide (BGT). Isolated mesenchymal cells (hMSC) from femurs of ovariectomized rats were differentiated into osteoblasts and submitted to in vitro tests. Bilateral ovariectomy (OVX) and sham ovariectomy (Sham) surgeries were performed in fifty-five female Wistar rats. After a period of 60 days, critical bone defects of 5.0 mm were created in the calvaria of these animals. For biomechanical evaluation, critical bone defects of 3.0 mm were performed in the tibias of some of these rats. The groups were divided into the clot (control) group, the BG group, and the BGT group. After the sonochemical process, the samples showed modified chemical topographic and morphological characteristics, indicating that the surface was chemically altered by the functionalization of the particles. The cell environment was conducive to cell adhesion and differentiation, and the BG and BGT groups did not show cytotoxicity. In addition, the experimental groups exhibited characteristics of new bone formation with the presence of bone tissue in both periods, with the BGT group and the OVX group statistically differing from the other groups (*p* < 0.05) in both periods. Local treatment with the drug teriparatide in ovariectomized animals promoted positive effects on bone tissue, and longitudinal studies should be carried out to provide additional information on the biological performance of the mutual action between the bioglass and the release of the drug teriparatide.

## 1. Introduction

Osteoporosis, considered a public health problem, is one of the most prevalent postmenopausal diseases in the world and is commonly associated with estrogen deficiency [1]. According to the International Osteoporosis Foundation, this disease is very common in the elderly, and one in three women over 50 will experience a fracture due to bone fragility during their lifetime [2]. This is because estrogen production is attenuated after the menopausal period [3], and this deficiency inhibits, mainly in late phases, the differentiation and maturation of osteoblasts, exerting a direct effect on these cells [4].

In this context, in patients suffering from osteoporosis, bone healing is a negatively affected process [5], making the treatment of bone defects or bone fractures in these compromised organisms challenging [6]. Therefore, due to changes in the normal bone remodeling process, it is sometimes necessary to implant biomaterials that can assist in the surgical treatment of patients with osteoporotic fractures. In these cases, one must consider not only the properties of the biomaterial but also the characteristics and regenerative capacity of the host bone [7].

Currently, autogenous bone is considered the gold standard in the replacement of bone defects [8], but the disadvantages associated with autogenous bone, such as the need for more than one surgical site, limited availability, and the possibility of the post-complications [9,10,11], have motivated the development of new biomaterials used in bone repair [12,13], particularly bioglass, which has excellent physical–chemical properties and a long history of applications as biomedical fillers, which has stimulated several researchers to test the use of these materials in tissue engineering and regeneration strategies [14,15,16,17,18,19,20,21]. Furthermore, the discovery of bioglass as the first artificial material with a clear ability to form an integrated bond with bones has stimulated the interest of scientists and clinicians for a long time [22].

Previous studies have shown that the formation of a surface layer of apatite hydroxycarbonate (HCA) after the ionic exchange between glass and body fluid is responsible for the bone binding mechanism of these materials [23,24], which also makes the role of the surface of bioglass important in bioactivity studies [25]. In tissue regeneration strategies, other materials such as metals, bioceramics, and biodegradable polymers can be associated with 45S5 Bioglass^®^ [26] increase the potential for application in tissue engineering [17], making this material attractive to be evaluated as a bone substitute in adverse situations.

The 45S5 Bioglass^®^ has great versatility in the fields of biomedicine, orthopedics, and even dentistry, being used for bone repairs in the fields of periodontics [27], maxillofacial surgery [28], and even for soft tissue repair [29]. Furthermore, the ability of bioglasses to incorporate hydrophilic and hydrophobic groups in their structures resulted in their development in association with therapeutic agents [30,31,32,33]. 

One of the new perspectives in the field of tissue engineering research is the use of biomaterials associated with drugs that promote a joint action to act on bone repair [31,34,35]. Teriparatide (PTH 1-34) is a drug analogous to parathyroid hormone (PTH 1-84), which, when administered systemically, proved to be effective in reducing the osteonecrotic area in the jaws of animals receiving doses of zoledronic acid, a drug from the group of bisphosphonates, and undergoing tooth extraction compared to the group that was not treated with teriparatide [36]. In addition, teriparatide stimulates cortical and trabecular bone formation, as well as increases bone strength and volume [37]; this is because this drug has a different mechanism of action than medications a currently available for the treatment of osteoporosis, with improved bone quantity and quality in osteopenic animals [38,39,40]. PTH-34 facilitates cell proliferation and differentiation, acting on cell angiogenesis and endothelial cell function as demonstrated by Jiang et al. [41].

Because of the association between bioglass particles and local medicines for osteoporosis in the in vivo response, the choice of this drug was due to its favorable effects on bone tissue metabolism [39,40,42,43] and evidence in the literature on its effect on cell cultures [44,45].

The drug incorporation process on the surface of biomaterials can occur in different ways; among them, sonochemical processing stands out [31] because in this technique, the ultrasound equipment generates the formation of bullous cavitations through radiation contact with the material [46], promoting changes in the physical and structural characteristics of the biomaterial, such as reduction, homogenization of particles, and occasionally the formation of a superficial amorphous layer [46,47], which is necessary for the incorporation of the drug. Gonzalo-Juan et al. [48] used the sonochemical technique to produce silver nanoparticles and incorporate them into the surface of the bioglass and proved that this technique does not interfere with the bioactivity mechanisms of glass, as it does not significantly change the network structure of the glass, considering this easy and fast route.

From this context, the investigation of new biomaterials that can accelerate or promote bone regeneration in patients who present osteoporosis and require surgical procedures is of paramount importance and has been addressed in current studies [15,16,17,18,19,20,21]. Therefore, the development of a biomaterial that favors the bone regeneration process and can positively contribute to the field of tissue bioengineering, promoting a combination of the advantages of teriparatide medication, which proves to be an innovative bone anabolic drug, with bioglass that has properties positive in relation to bone repair becomes of extreme necessity.

## 2. Materials and Methods

### 2.1. 45S5 Synthesis

We used the method described by Spirandeli et al. [49] to obtain 45S5 bioglass, which is based on the processes of preparing glass ceramics by sol-gel followed by melting–quenching. The fusion of the 45S5 bioglass was carried out in an oven at 1350 °C/15 min, in a ZAS crucible (zirconia-alumina-silicon). Subsequently, the molten glass was poured into water. The glass, in the shape of fries, was collected, dried, and then ground in a mechanical mortar for 3 h and later sieved through a 200 mesh sieve. The composition of the 45S5 bioglass, in % mol, was 46.1% SiO_2_, 24.4% Na_2_O, 26.9% CaO e 2.6% P_2_O_5_.

### 2.2. Sonochemical Technique

All samples were subjected to ultrasonic processing carried out at the Advanced Materials Laboratory of the Center for Research and Development of Functional Materials—Health Division—CEPID/FAPESP in a Sonics VCX-750 (SONICS Vibra Cell™, Newtown, EUA) brand model, with 750 W power and 20 kHz frequency, with 5 min pulses and variable amplitude up to 70% of the Sonics equipment nominal amplitude (450 W/cm^2^). The 45S5 bioglass samples were divided into equal parts, one part consisting of functionalized bioglass in the absence of the drug, which was called group BG, and the other part consisting of bioglass subjected to functionalization processing associated with the drug teriparatide, which was identified as group BGT. The relative concentration in mass was 55 g/L determined in the initial stages of the process, using the hormone PTH 1-34 (Lilly France S.A.S, Fegersheim, France), 45S5 bioglass in the form of powder, and Milli-Q water.

### 2.3. Characterization Biomaterial

The surface topography of the samples was analyzed by the scanning electron microscope (SEM) (JEOL/JSM-5310, Tokyo, Japan) with “Field Emission Gun” (FEG) was used (Tescan/Vega 3, Brno, Czech Republic) before and after the functionalization of the samples. The images were obtained on the SEM with a secondary electron (ES) detector, projected on the sample surface. The original magnifications used were 10,000×, 20,000×, 50,000×. The functional groups on the surface of the particles were identified by analyzing them using the Fourier transform infrared spectrophotometry (FTIR) (Parkin Elmer, model Spectrum GX) in UATR mode, installed at the Associated Laboratory of Sensors and Materials of the National Institute for Space Research (LAS/INPE), in the middle region 500–4000 cm^−1^, 32 scans and 4 cm^−1^ resolutions using the Spectrom Search Plus program (PerkinElmer, Waltham, MA, USA). Energy-dispersive analysis was performed to map the elements using Bruker Nano GmbH 410, (Berlin, Germany) and Espirit 1.9 software (Bruker, Berlin, Germany) associated with SEM (Inspect S50, FEI Company, Brno, Czech Republic). The values of the Zeta potential of the bioglass were obtained using the dynamic light scattering equipment, Stabino Control 2.00.23, (Particle Metrix GmbH, Meerbusch, Germany) installed at the Research and Development Institute of the University of Vale do Paraíba (UNIVAP). The bioglass powders were dispersed in deionized water (pH 5.0), and the measurements were performed at 25 °C at an angle of 15°. To calculate the zeta potential from the mobility values, the Smoluchowski equation was used, with values of the refractive index, dielectric constant, and water viscosity at 25 °C.

### 2.4. In Vitro Experiment 

The mesenquimals cells (hMSC) were obtained from the femurs of nine ovariectomized female Wistar rats (Rattus norvegicus) as previously described by Zhang et al. [50]. After cleaning the femurs, in the laminar flow, the bone marrow cells were isolated and inserted in cell culture flasks of 250 mL and 75 cm^2^ (TPP, Biosystems, Curitiba, Brazil) with essential alpha MEM medium culture (Gibco) supplemented with 10% Bovine Fetal Serum (SBF) (LGC Tenchology, Campinas, Brazil) and gentamicin (500 μg/mL) (Gibco) and were incubated in an oven at 37 °C, with atmospheric humidity containing 5% carbon dioxide (CO_2_). The culture medium was changed every three days, and the progression of the culture was evaluated by inverted phase microscopy (Microscope Carl Zeiss—Axiovert 40C, Oberkochen, Germany). After confluence (approximately seven days), cells were released enzymatically [51] and plated at a density of 1 × 10^4^ viable cells in each well of the 96-well microplate (Transwell, Corning/Costar, New York, NY, USA).

Before plating, samples were weighed on a semi-analytical precision scale (BEL Engin Mark, model 210A) at 0.022 g. Subsequently, these samples were sterilized in UV light and placed inside the wells. In the wells of the control group, only cells were plated. Then, osteogenic culture medium was added to the plate. The final volume of the osteogenic medium was 250 μL per well, which was changed every 48 h.

After these procedures, all plates were incubated at 37 °C with 5% CO_2_ and kept until the tests. All tests were performed according to ISO 10993-5 [52] and in triplicate, with each isolation being a pool of cells from the femurs of three animals.

#### 2.4.1. Cell Adhesion

After 3 and 5 days of cultivation, cell morphology was evaluated by FE-MEV (Field Emission Scanning Electron Microscopy) (Zeiss—EVO MA10, São Paulo, Brazil) at the Dental Materials and Prosthesis Research Laboratory of ICT/Unesp. The samples were coated with a thin layer of gold using a sputter-coating system. The samples were placed on the stub (aluminum platform), aided by a double-sided carbon tape (3M, Sumaré, Brazil), and metallized with a thin layer of gold by sputtering in the metallizing machine (EMITECH K550X, Sputter Coater, Quorum Technologies, Lewes, UK) for a period of 130 s. The images were obtained by SEM with a secondary electron detector in several magnifications.

#### 2.4.2. Cell Viability (MTT)

After the periods of 3 and 7 days, a quantitative assessment of live cells was performed, after exposure to the toxic agent by incubation with the MTT [3-4,5-dimethylthiazole bromide] (Sigma Aldrich, St. Louis, MO, USA), and the formazan crystals were dissolved by adding 100 μL of dimethyl sulfoxide solvent (Sigma-Aldrich) to each well. The plates were shaken at room temperature to dissolve the crystals, and the absorbance was measured spectrophotometrically at 570 nm (Micronal AJX 1900). Results are expressed accepting the absorbance of the negative control as indicating 100% viability.

#### 2.4.3. Protein Content Determination and ALP Assays during hMSC Differentiation

The total protein content was calculated in two different periods, 3 and 7 days, to assess whether the biomaterial accelerates the production of cellular proteins or not. This measurement was performed according to the modified method of Lowry [53]. The absorbance was measured spectrophotometrically at 680 nm (Micronal AJX 1900). The alkaline phosphatase (ALP) activity was determined by releasing thymolphthalein by hydrolysis of the thymolphthalein substrate, using a commercial kit according to the manufacturer’s instructions (Labtest Diagnostic), in the same periods of the total protein, using the same lysates. The absorbance was measured in a spectrophotometer (Micronal AJX 1900) at 590 nm.

#### 2.4.4. Formation of Mineralization Nodules 

After 12 days of culture by staining Alizarin S2% red (Sigma-Aldrich, St. Louis, Brazil), pH was measured to be 4.2. Specifically for this test, microplates with the presence of a Transwell net (Transwell, Corning/Costar, New York, NY, USA) were used. The red dye from Alizarin S was used to stain areas that are rich in calcium. This test was used to assess whether there was an acceleration in the production and calcification process of the cellular matrix and to characterize mesenchymal stem cells as osteoblasts due to the production of mineralized matrix. The quantification of calcium in mineralized formations was performed according to the method described by Gregory et al. [54]. The absorbance was measured in a spectrophotometer (Micronal AJX 1900) at 405 nm.

### 2.5. Experimental Design In Vivo Study

This work was executed according to the ethical principles of experimentation (CONCEA) and approved by the local ethic committee (Processes numbers 10/2019) adopted by the National Council of Control of Animal, and ARRIVE [55] was respected. Forty-five female Wistar rats (Rattus norvegicus) (Central Vivarium of Unesp Botucatu, São Paulo, Brazil) that were approximately 90 days old and 350 g in weight were used and kept in cages with water ad libitum and diet. The animal was randomly divided into two groups, with 30 rats having undergone bilateral ovariectomy (OVX) and 15 rats having undergone simulated ovariectomy surgery (Sham). Thus, the experimental model adopted in the present project consisted of estrogen-deficient animals in the OVX group [18,56,57,58]. In each OVX and Sham group, critical 5.0 mm bone defects were made in the calvaries. For biomechanical evaluation, OVX rats (*n* = 10) and Sham rats (*n* = 5) underwent 3.0 mm bone defects on the right and left tibiae. All defects on the left side in OVX group were filled with biomaterial—(a) functionalized bioglass (BG) and (b) functionalized bioglass with 10% teriparatide (BGT)—and the right sides were filled with clot. All defects on the sham group were filled with clot. This distribution aimed to avoid the possible influence of one material on the other resulting in 5 rats for each subgroup according to the period. After 2 and 6 weeks, the animals were euthanized and the bone repair area was evaluated through histological, histomorphometry, immunohistochemical, and biomechanical tests.

#### 2.5.1. Bilateral Ovariectomy

For the induction of osteoporosis, forty rats were subjected to bilateral ovariectomy (OVX). The rats were anesthetized with xylazine hydrochloride (Xylazine—Coopers, Brasil, Ltd., Osasco, Brazil) and ketamine hydrochloride (Injectable Ketamine Hydrochloride, Fort Dodge, Health Animal Ltd., Osaka, Japan) Additionally, they were then placed in lateral decubitus to perform an incision of 1 cm (cm) on the flanks and the subcutaneous tissue, and then the peritoneum was divided into planes to access the abdominal cavity. Subsequently, the ovaries and uterine horns were located, which were cauterized using the cauterizer (Cautermax-Fabinject^®^). Subsequently, the sutures were made in layers, all using silk thread no3 (Ethicon/Johnson & Johnson, Blue Ash, OH, USA). Twenty rats in the sham surgery group (Sham) underwent the same procedure, but only the surgical exposure of the uterine horns and ovaries was performed without their respective cauterizations. 

#### 2.5.2. Surgical Procedure at Calvaria

After 60 days of the ovariectomy (OVX) and simulated ovariectomy (Sham) procedures, the same rats were subjected to the manufacture of critical bone defects in the calvaria. The animals were weighed and anesthetized with a solution of xylazine hydrochloride 2.3 g/100 mL (Anasedan^®^, Vetbrands, Jacareí, Brazil) and ketamine hydrochloride 1.16 g/10 mL (Dopalen^®^, Vetbrands, Jacareí, Brazil).

After the anesthesia of the animals, the surgical sites were subjected to trichotomy and antisepsis with an iodized alcohol solution. A linear incision of approximately 3.5 mm was performed with a scalpel blade number 15 in the region corresponding to the medial face of the calvaria. The tissues were divulsed to expose the calvaria cortices, in which the 5.0 mm defects were made under abundant and continuous irrigation with physiological solution to prevent heating due to the friction of the drill with the bone. To obtain the critical defect, a 5.0 mm diameter drill was used. The defects on the left side in the OVX group were made with (a) bioglass (BG) or (b) bioglass associated with 10% teriparatide (BGT), and on right side, they were filled with clot. In the sham group, the defect was filled only by clot. For the stabilization of the material in the surgical site of the defect, a larger-diameter biological membrane (GenDerm^®^) was positioned at the bone defect site. In all defects, the tissues were repositioned, and the layers were sutured with no3 silk thread (Ethicon/Johnson & Johnson). Again, iodized alcohol antisepsis was performed. After surgery, the rats were placed in mini-isolate of the ventilated rack and monitored until the euthanasia deadline.

#### 2.5.3. Macroscopic Evaluation of the Uterus and Exfoliative Cytology

Euthanasia was performed within 2 and 6 weeks with an overdose of the drugs used for anesthesia, administered intramuscularly. Overdoses 4 times greater were used to ensure that the animal did not return consciousness. In addition, to prove the effect of ovariectomy, the macroscopic aspect of the uterine horns of the rats of the Sham group and of the rats of the OVX group was observed, and the estrous cycle was assessed.

#### 2.5.4. Histological and Histomorphometric Analysis

The calvaries were removed, cleaned, and placed in a 10% formaldehyde solution for 48 h. Then, the samples were placed in a solution of ethylenediaminetetraacetic acid (EDTA Titriplex III, EMD Millipore, Burlington, MA, USA) for demineralization for about 60 days. The parts were checked regularly, and when demineralization was found, the pieces were sectioned longitudinally in the center of the bone defect of the calvaria, inserted in plastic cassettes, and placed in a tissue processor (Leica TP1020, Wetzlar, Germany) for later inclusion in paraffin. Then, semi-serial sections 5.0 μm (µm) thick were obtained, which were stained with hematoxylin and eosin (HE), and 3.5 µm sections that were extended on silanized slides for immunohistochemistry analyses. In the descriptive histological analysis, aspects of the development of bone repair were observed to evaluate the formation of granulation tissue, bone neoformation, the arrangement of immature bone trabeculae, and bone maturation until final remodeling. The morphometric analysis of the bone neoformation was realized with five slices from each animal stained with hematoxylin and eosin. Stained sections were examined by light microscopy under 5× objective lenses, and images were obtained with a Microscope Zeiss Axioskop 40 (Carl Zeiss) and analyzed with software Image J (Image Processing and Analysis in Java, NIH, Bethesda, MD, USA) by two calibrated and blinded examiners using a light microscope.

#### 2.5.5. Immunohistochemistry Analysis

The slides were prepared for immunohistochemical reactions by blocking endogenous tissue peroxidase by incubation with 6% hydrogen peroxide and methanol. Antigenic recovery for osteocalcin was performed by heating it in a glass vat containing 10 mMph 6.0 citric acid in the microwave (700 w) for 2 min; after heating, the slides were placed in the vat with citric acid heated for 40 min. Then, the primary antibodies were incubated, which occurred according to the dilution at 1:100 overnight and at a temperature of 4 °C. Subsequently, the incubation with the secondary antibody (Universal LSAB TM Kit/HRP, Rb/Mo/Goat—DAKO) was carried out for 30 min at room temperature, the biotinylated anti-goat secondary antibody produced in donkeys (Jackson Immunoresearch Laboratories) was used, and the amplifiers selected were avidin and biotin (Vector Laboratories, Burlingame, CA, USA). The chromogen used was the diaminobenzidine solution (Dako, Glostrup, Denmark). Subsequently, the cuts were stained with Mayer’s hematoxylin, dehydrated in ethanol, bleached in xylol, and mounted with Permount. Negative controls were performed by replacing the primary antibody with bovine serum albumin (BSA), and positive controls were performed as suggested by the manufacturers of the primary antibodies.

For the analysis, a standardized photo of each slide was performed, with a ×40 magnification of the region of the interface between the bone defect and the old bone, to observe the expression of the osteocalcin (OC) and RankL. Evaluations were performed on the edges of the defects in both periods by means of visual evaluation and were made by the same evaluator in blind conditions. For qualitative analysis, immunostaining for cells was considered [59,60,61,62,63]: (a) (−) = negative; (b) (+) = positive; (c) (++) = superpositive; (d) (+++) = hyperpositive. After the qualitative analysis, semi-quantitative analysis was carried out by converting the scores into percentages: 0% to negative, 20% to positive, 60% to superpositive, and 90% to hyperpositive. A higher percentage reflected an increase in the number of cells stained positively by diaminobenzidine in the area.

#### 2.5.6. Surgical Procedure at Tibiae

After 60 days of the ovariectomy and sham procedure, 20 rats were again anesthetized using the same technique and submitted to the surgical procedure for making the critical defect. With a carbide bur, a critical size defect of 3.0 mm in diameter was performed under copious irrigation with 0.9% sodium chloride. Critical bone defects in the left tibiae of OVX rats were filled with BG or BGT. On the right side, clot stabilization (control) was expected. The Sham group was filled with clot. The flap was repositioned and sutured with #4 silk thread (Ethicon/Johnson & Johnson). Two weeks after surgery, the animals were euthanized, and the tibiae were submitted to a three-point flexion test to verify the influence of the biomaterial on the mechanical property of the newly formed bone tissue.

#### 2.5.7. Biomechanical Properties

After euthanasia, the tibiae were kept in Ringer’s solution at –20 °C, until the moment in which the analyzes were performed in the Research Laboratory in Dental Materials and Prosthesis of the ICT/Unesp. To carry out the test, each specimen was placed centrally, along its length, on a support containing two supports (15 mm from each other), with its anterior face facing downwards. The load was applied transversely along the long axis of the tibia on its posterior face at a midpoint between the two supports. The load application support and the supports used had a cylindrical shape with a diameter of 3 mm. The test was conducted on a universal testing machine (Emic^®^—model DL 200 MF, Testing Equipment and Systems Ltd., São José dos Pinhais, Brazil), which provided a force of 50 kg/F with a constant application speed of 5.08 mm/min until specimen failure.

### 2.6. Statistical Analysis

All statistically analyzed tests were submitted to the Kolmogorov–Smirnov normality test, and homogeneity of the results was observed (*p* > 0.05). To analyze the Zeta potential in the characterization of the material, the data were analyzed by one-way ANOVA (*p* < 0.05). Data from in vitro tests and histomorphometry and immunohistochemical analysis were analyzed by two-factor ANOVA (*p* < 0.05), considering the period and biomaterial analyzed as variables, and the data are presented as mean value ± standard deviation. The biomechanical evaluation was carried out within only 2 weeks, so it was submitted to one-way ANOVA (*p* < 0.05). When necessary, they were submitted to Tukey’s post hoc test. All tests were performed using the GraphPad Prism 6.0 software (GraphPad Software, San Diego, CA, USA) and for all statistical tests, a significance level of 5% and a power of 80% were adopted.

## 3. Results

### 3.1. Sample Characterization

Based on Figure 1, it was possible to observe that the surface of the 45S5 bioglass without functionalization presented particles of geometric shapes and right angles compatible with the morphological characteristics of glassy materials (Figure 1(A1,B1). The agglomeration of the glass particles could also be observed due to less surface energy. However, the functionalized bioglass (BG) and functionalized bioglass groups were associated with 10% teriparatide (BGT). Additionally, Figure 1(A2,A3,B2,B3) show particles with rounded angles and more solubilized, suggesting that the surface was attacked chemically by the process of functionalization. 

Figure 2 shows three FTIR spectra corresponding to the 45S5 bioglass, functionalized bioglass (BG), and functionalized bioglass groups associated with 10% teriparatide (BGT). The sample composition before and after the functionalization of the bioglass in the absence of the drug has chemical bonds that formed networks characteristic of the bioglass. It is then possible to confirm, by means of the infrared spectroscopy technique, the presence of bands that represent the bonding groups of BO (Si-O-Si), phosphate (PO_4_), and carbonate (CO_3_) bonds and non-oxygen bridge bonds NBO (SiO-Ca^2+^ O-Si and SiO-Na^+^ O-Si) in all groups. It was observed that there was a small reduction in the relative intensity of the NBO bands in the functionalized material in the presence of the drug teriparatide (BGT) when compared to the other samples and the presence of bands in the amide group (C=O) consisting of bonds between carbon and hydrogen (CH); nitrogen and hydrogen (NH); and carbon, oxygen, and nitrogen (CON) (1660 cm^−1^) in this group. The changes in the control of bands around 1000 cm^−1^ and 1500 cm^−1^ occurred by overlapping the medication bands with those of the bioglass. The carbonate group (CO^3−^) showed no significant change in any of the groups.

The samples, when evaluated by EDS, show the presence of chemical elements that make up the 45S5 bioglass. Mainly, energy peaks characteristic of the elements calcium (Ca), silicon (Si), and oxygen (O) were observed in the bioglass 45S5 and BG groups (Table 1), as well as the presence of Ca and P in all groups. Meanwhile, samples that were functionalized with the drug teriparatide exhibited sodium peaks (Na) as the main element and decreased Si element. Furthermore, the presence of the chemical element gold (Au) was not observed in the particles of the 45S5 bioglass group. Particles in the graph in Table 1 the chemical element gold (Au) were not observed.

All samples in Zeta potential test demonstrated the surface reactivity of the particles. The values obtained in the functionalized bioglass particles with the presence of 10% teriparatide (BGT) showed a higher magnitude (*p* < 0.05; One-way ANOVA, Tukey’s test) (Figure 3). The 45S5 bioglass showed fewer functional groups, whereas the functionalized groups had an equivalent number of functional groups, since there was no statistical difference in the samples.

### 3.2. In Vitro Analysis

After 3 and 5 days of culture, the samples were analyzed by scanning electron microscopy (SEM) to show the cellular interaction on the materials (Figure 4). In this analysis, it was evidenced that all the samples allowed cell spreading. Although the materials exhibited irregular macrostructures and reveal a rough surface with many pores and reduced coalescence between the particles, it was possible to observe cellular extensions permeating the particles of the material.

In the period of 3 and 7 days, the groups did not show statistically significant differences (*p* > 0.05). When performing the analysis between the periods, it was verified that the BG and BGT groups in the 7-day period differed statistically from the BG group and the control group in the 3-day period (*p* < 0.05), presenting the cell with the highest viability value. The results are shown in Figure 5A.

Within the evaluated periods, cellular metabolic activity was verified in all groups. In both periods, the mean value of the total protein amount was higher in the bioglass group associated with the drug teriparatide 10% (BGT), but within 3 days, there were no statistically significant differences (*p* > 0.05) between the groups analyzed. In the 7-day period, there was also no statistical difference between the BGT and the BG group (*p* > 0.05), but there was statistical difference with control group (*p* < 0.05). It was possible to observe that all experimental groups (BG and BGT) had a statistically significant increase in the largest period analyzed when compared to the period of 3 days (*p* < 0.05), while this fact is not observed in control group. The results are shown in Figure 5B.

In the same lysates from the analysis of total protein content, ALP was measured. In the period of 3 days, the activity of ALP showed no statistical difference from the other groups. It was found that in the 7-day period, the experimental groups showed higher mean values of ALP when compared with the same groups in the period of 3 days, but it also did not differ statistically (*p* > 0.05). Between the periods evaluated, there were no statistically significant differences between any groups (*p* > 0.05). The results are shown in Figure 5C.

After 12 days of cell culture, the formation of mineralization nodules was observed in the experimental groups. It was observed that the quantification of calcium in nodule formation (Figure 6) was greater in the functionalized bioglass group (BG), differing statistically from the control group and BGT group (*p* < 0.05). Figure 6 represents the nodules formed in the wells of the experimental groups.

### 3.3. Bone Repair Analysis

In the 2-week postoperative period, there was no evidence of inflammation or infection at the surgical site. Microscopically, the pieces evaluated had longitudinal sections of the calvaria of the rats where the biomaterials were positioned. The extremities of the surgical sites contained newformed bone tissue showing bulky osteocytes and connective tissue with the presence of spindle cells. The analyzed period of 2 weeks showed signs of bone tissue remodeling with the proliferation of bone trabeculae from the ends of the bone cortex, close to the region where the material was positioned (Figure 7). This bone showed bulky osteocytes within wide gaps and osteoblasts, arranged in rows around the disorganized trabeculae, and it was also possible to verify the presence of the biomaterial in the areas that contained thicker connective tissue. Specifically, in the experimental groups incorporated with the drug teriparatide (BGTO), the area of the bone defect stands out, with the presence of a thicker connective tissue showing a large amount of blood vessels, while in the other groups, the presence of loose connective tissue and little inflammatory infiltrate (Figure 8).

In the period of 6 weeks, it was possible to notice that there was still no closure of the defect with bone tissue in any evaluated group; in addition, it was observed that a large part of the biomaterial had already been reabsorbed, with the concentration of bone formed in the stump areas. In these areas, more mature bone tissue was present, with evident Havers canals and lamellae that contained osteocytes within gaps, also arranged in concentric rings. It is possible to observe the presence of compact bone next to the residual biomaterial. The connective tissue is denser and there is a large amount of thicker collagen fibers when compared with groups in the period of 2 weeks. Based on data from descriptive statistics, when considering the influence of the critical-defect filling materials and the periods, the ANOVA-two-factor test of variance was performed, followed by the Tukey multiple comparison test. The results comparing the measured values of the newly formed area between the groups and periods analyzed are shown in Figure 8A. Therefore, when the analysis among the groups from the same period occurred, it was concluded that in both the period of 2 weeks and 6 weeks, there were no statistically significant differences (*p* > 0.05), while when comparing the periods analyzed, statistically significant differences (*p* < 0.05) were observed between all groups in the 2-week period with the BGTO6 group.

The highest maximum force values were observed in the BGTO group, that is, ovariectomized animals with bone defect filled with bioglass functionalized with teriparatide. This group exhibited a statistically significant difference from the other groups (*p* < 0.05). All other groups did not differ from each other (*p* > 0.05) (Figure 9B).

Immunomarking of osteocalcin was concentrated at the edges of the defects in both periods. In all analyzed groups, osteocalcin immnunomarking was observed next to the mineralized bone matrix, being more observed in the later period (6 weeks) of bone neoformation (Figure 10 and Figure 11).

The average values of cells immunomarked in percentage for osteocalcin demonstrated that in the period of 2 weeks, there were no statistically significant differences (*p* > 0.05), but the Sham and OVX clot groups showed higher mean values than other groups. However, when observing the period of 6 weeks, the BGTO group showed a clear increase in cells immunomarked by osteocalcin compared to the mean values of the period of 2 weeks, but without showing statistical differences (*p* > 0.05). These results can be observed in Figure 11C.

In the qualitative analysis it was possible to observe that osteocalcin (OC) was considered superpositive in the CS groups and positive in the CO group within 2 weeks, with marking on the newly formed bone trabeculae, osteocytes, and in the extracellular matrix. Meanwhile, the experimental groups expressed positive and sometimes negative markings for osteocalcin (OC). The osteocalcin marking pattern indicated that the bone tissue in the period of 2 weeks in the BGO, BGTO groups was immature when compared to the clot groups (Figure 10). In the 6-week period, it was observed that the clot groups showed superpositive marking for osteoblastic cells and osteocytes in the matrix. The same was true of the experimental groups that were superpositive and positive, presenting areas of the extracellular matrix expressing osteocalcin, as well as osteoblasts and osteocytes (Figure 11).

In the qualitative analysis, within 2 weeks, the RankL immunostaining in the nucleus of newly trapped osteocytes in the bone matrix neoformed in osteoblasts in the periphery of the neoformed bone and in fibroblasts present in the connective tissue (Table 2). It was possible to observe that the immunostaining of cells for RankL was considered superpositive in the CO e BGTO group for a 2-week period, while the CS and BGO groups showed less evident markings. It was observed that CO showed a reduction in immunomarked cells in the connective tissue (Figure 12). Within 6 weeks, the presence of more mature bone tissue can be observed, with smaller gaps of osteocytes presenting immunostaining in osteoblasts present in the osteoblastic rhyme of the newly formed bone, as well as in some connective tissue cells for all groups, presenting for all experimental groups and the CO-positive immunostaining group on the RankL scale (Figure 12). On the other hand, the CS group within 6 weeks showed hyperpositive immunostaining for RankL, but more evident markings were observed in the connective tissue and little marking in osteocytes and osteoblasts was present in the matrix and in the periphery of the newly formed bone, respectively (Figure 13). The result of the semi-quantitative analysis of the RankL biomarker showed that the CO group had a higher mean value of immunomarked cells as a percentage in the period of 2 weeks, however, without showing statistically significant differences (*p* > 0.05) with the other groups. In the period of 6 weeks, again, there was no statistically significant difference between the groups (*p* > 0.05), but the CS group had the highest mean value of immunomarked cells, while the other groups showed a decrease in the number of cells marked for RankL in this period (Figure 13).

## 4. Discussion

One of the most prevalent postmenopausal diseases in the world is osteoporosis, a public health problem that sometimes occurs due to estrogen deficiency [64]. However, although it is known that osteoporosis causes an increase in bone loss, the specific mechanism of action of estrogen in bone tissue remains unclear [45]. In cases of individuals who have chronic diseases associated with the need for bone repair, if necessary, it can be difficult [65,66]. Therefore, there is a growing demand for biomaterials that have adequate properties in relation to bone tissue and that can positively influence tissue regeneration, improving biomechanical properties and accelerating the osteogenesis process [31,32,67,68].

One of the biggest challenges for the treatment of injuries resulting from osteoporosis is the low availability of drugs with local action. In this context, recent advances have encouraged the association of biomaterials and drugs as a local drug-release strategy [31]. According to Kyllonen et al. [69] for the release and transport of the active drug, particles of different sizes can be used; therefore, the drug is incorporated, absorbed, or associated with these particles and thus is protected from the degradation from the environment until its release. In a recent study by Mosqueira et al. [32], the authors concluded that the efficiency of the loading of therapeutic agents in strontium-modified bioglass spheres, as well as the drug release rates, were mainly influenced by the pore size of the particles, thus improving the osteogenic potential of the mesenchymal cells of the bone marrow obtained from rats with osteoporosis. In this study, bioglass particles were reduced to nanometer size for surface increase, as the number of surface atoms increases dramatically as the particle size decreases [70], which contributes to drug incorporation and release. In a study by Ajita et al. [71] the authors demonstrated that nanostructured bioglass particles (37.6 ± 0.81 nm) had applications in bone treatments, as well as materials that form the link between bone tissue, showing greater cell proliferation of mesenchymal cells from mice in bioglass dissolution products with smaller particles.

Bioglasses are interesting biomaterials for use in bone repair, as they can stimulate osteogenesis and angiogenesis [29,72,73], improving the adhesion, proliferation, and differentiation of mesenchymal stem cells from stem cells into osteoprogenitor cells and increasing the mineralization rate [74]. Furthermore, the main advantage of bioglass is its ability to bond to bone tissue due to the formation of an interfacial layer of calcium phosphate [22,75,76,77,78] As the understanding of the bioglass structure is directly associated with the interaction mechanisms of the material with the fabric, the structural investigation of the particles was carried out through several characterization tests of the samples. SEM-FEG results showed changes in the physical characteristics of the particles, in which the functionalized groups had rounded edges and reduced sizes in relation to the bioglass sample before the sonication process, changes that were characteristic of the functionalization process [56]. The functionalization time of the sample is of paramount importance, as the increase in the sonication period of the bioglass promotes a decrease in the size of the particles and smoothing of the edges of the sample. The sonochemical technique was recently used by the same research group in the evaluation of the local release of PTH 1–34 functionalized to Biogran^®^ in peri-implant bone regeneration in osteoporotic conditions. The results obtained were promising for the osteoporotic group associated with the drug compared to the other groups, with the peri-implant repair for osteoporotic bone functionalized with PTH 1-34 similar to that of healthy bone [76].

While the FT-IR analyses indicated great structural similarity between the 45S5 bioglass and BG groups, similar to the deconvolutions of the FTIR spectra in the study by Lee et al. [77], who showed peaks associated with the structure of the bioglass, the BGT group was different, showing bands from the standard amide group (1660 cm^−1^) of the teriparatide drug molecule. This same band was observed in the study by Bahari Javan et al. [79]. A decrease in the band of the group not bound to oxygen (NBO) can be observed; these bands are necessary for the dissolution of the bioglass, which will result in the formation of a layer of apatite hydroxycarbonate (HCA) on the surface of the material when in contact with fluids body or aqueous media. It has been suggested that the decrease in the number of NBO bonds is due to the hydrolysis of the bioglass, a process that leads to the formation of a silica network in the topographic area of the material, as described by Fiume et al. [80]. The condensation stage, at this moment of the sonochemical process, incorporates the drug teriparatide on the surface of the biomaterial, which is confirmed by the presence of bands of amide groups in this group. These observed changes may also be indicators of the incorporation of drugs in the silanol groups (Si-OH) of the bioglass [31].

When analyzing the EDS results, it was possible to verify that all groups presented the presence of Ca and P in these samples (Table 1), suggesting the presence of the formation of calcium phosphate phases on the surface of the material, which did not suffer from alterations in the presence of the drug teriparatide [81]. The electrical charge and protein adsorption of the material are also closely associated with the understanding of the biological interaction mechanisms of biomaterials with the tissue, so the analysis of the zeta potential was performed with the objective of obtaining the surface electrical energy properties of the biomaterial, where the zeta potential is influenced according to the medium in which the particles interact. The greater the value of the zeta potential is in magnitude, the greater the stability of the particles in the medium [82]. In this study, the 45S5 bioglass and the functionalized groups exhibited surfaces with different surface charge properties; it was possible through this test to observe that the functionalized groups presented greater amounts of functional groups when compared to the 45S5 bioglass, and this characteristic provides a favorable surface to cell adhesion that can cause greater protein adsorption [82,83].

Furthermore, for the development of a new biomaterial to be used in tissue regeneration, it must be ensured that it does not present potential toxic effects when implanted in the body and stimulates cell activity and differentiation. In the present study, the bioglass was functionalized by the ultrasonic sonochemical technique associated or not with the drug teriparatide 10% and used to verify the behavior of osteoblasts when in contact with samples of these biomaterials through cytotoxicity tests (MTT), total protein content, alkaline phosphatase activity, cell adhesion, and the formation of mineralization nodules, verifying the influence of these materials on osteoblastic cells.

Cytotoxicity tests indicate the effects of samples on the viability of cultured cells and is a predictor of potential toxic or non-toxic effects of biomaterials [84]. The results of the MTT test showed that all samples were biocompatible with values above 70% for viable cells in the two analyzed periods. In the present study, in the longest period of cell culture, the experimental groups exhibited more attenuated means of cell viability, as observed by Westhauser et al. [85], in which they evaluated the 45S5 bioglass associated with scaffolds in mesenchymal cells (hMSC) and observed a decrease in cell viability over longer periods. When the experimental groups were compared to the control group in relation to the total protein content, it was possible to observe that the BG and BGT groups had higher mean values in both periods; when statistically evaluating the groups and periods, it was possible to verify that the BG and BGT in the period of 7 days showed statistical differences with the control group of the same period and all groups in the period of 3 days (*p* < 0.05). This result is plausible, since the synthesis of several proteins that make up the extracellular matrix is influenced by the formation of a superficial layer of hydroxyapatite and silica present in the bioglass, which is responsible for cellular bioactivity [86]. This is because carbonated hydroxyapatite (HCA) is a bioglass dissolution product that positively interferes with the production of extracellular proteins, while BGT, in addition to presenting the bioactivity characteristics of bioglass, also presents the action of the drug teriparatide (PTH 1-34), for which in previous studies it was demonstrated that its local release acts positively on the influence of osteoblastic genes [39,87]. The expression of alkaline phosphatase (ALP) showed no statistically significant intragroup differences (*p* > 0.05) regardless of the period analyzed. ALP is important as it is one of the most reliable markers for osteogenic differentiation [88]. Tsigkou et al. [89] demonstrated that embryonic stem cells of the osteoblastic lineage in vitro start production and maturation after their cell differentiation, expressing several important proteins for the formation of the extracellular matrix, such as extracellular proteins, among them alkaline phosphatase (ALP), sialoproteins, collagenase I, osteopontin, and osteocalcin. The bioactivity of the drug teriparatide on the ALP activity of hMSCs indicated an increase in osteogenic differentiation of hMSCs and stimulation of osteoblastic cell differentiation, both in the continuous release of teriparatide and the pulsatile stimulation of the liberation of this drug [90].

Confirmation of the differentiation of mesenchymal cells (hMSCs) into osteoblastic cells in samples from the experimental groups and the control group was performed by the formation of mineralization nodules, as also observed by Rodrigues et al. [91]. This test is used to assess whether there was acceleration in the process of production and calcification of the cell matrix and characterizes the mesenchymal stem cells as osteoblasts, due to the production of a mineralized matrix [88,92]. In the present study, the BG group presented higher mean values, statistically differing from the other groups (*p* < 0.05). The result of the BGT group may suggest that due to the action of the drug teriparatide, a parathyroid hormone involving specific cellular receptors in osteoblasts and signaling pathways [93], the reproduction of studies only in vitro did not favor the observation of this action of the drug.

Therefore, for a better investigation of the ability to repair the critical defect in calvaria filled with 45S5 bioglass functionalized with 10% teriparatide and considering that the biomaterial can act positively on bone repair in conditions of osteoporotic bone injury, an in vivo analysis was performed. The ovariectomy procedure was performed to suppress the hormone estrogen in these animals. This methodology is already established and results in increased bone loss [43,58,94,95]. This result is due to the production of inflammatory cytokines, such as tumor necrosis factor alpha (TNF-α) from T cells and the production of osteoclastic cytokines, which induce the uncoupling of bone formation in the face of resorption; that is, bone formation does not follow the same pattern, i.e., the abundant rhythm of its reabsorption [96,97]. In in vivo studies, the success of ovariectomy can be confirmed by changes in the animal’s estrous cycle [98], using exfoliative cytology as a method of detecting changes in cell morphology during this cycle in the ovariectomized animal [57,99]. In this study, the estrous cycle was evaluated, and it was possible to observe in the vaginal smears that all ovariectomized rats remained in the diestrus phase, confirming the hormonal alteration. According to Luvizuto et al. [58], the permanence of the animal in this phase characterizes the presence of a hormonal state compatible with hypoestrogenism.

In the present study, in the descriptive histological analysis in the period of 2 weeks, the closure of the critical defect was not observed in any group; it was also possible to observe the presence of few inflammatory cells, and in the animals that received the BGT group, there was a large amount of granulation tissue, suggesting that the bioglass dissolution products and the influence of teriparatide on the critical defect were able to increase cell recruitment [17]. For confirmation of cell recruitment, the evaluation was performed at 6 weeks. However, for a material to become promising to be indicated for tissue regeneration, it is expected that after this recruitment, there will be a gradual degradation of the biomaterial with replacement by the original tissue [100]. In the period of 2 weeks, the presence of newly formed bone tissue also occurred mainly in the regions of the stump of the defect, observing many blood vessels and young fibroblasts in the experimental groups. These results agree with the previous study by Day et al. [101], in which the authors reported that 45S5 Bioglass^®^ promoted increased secretion of vascular endothelial growth factor (VEGF) in vitro and increased vascularization in vivo, suggesting that scaffolds containing Bioglass^®^ can stimulate neovascularization, generating beneficial effects for large tissue-engineering constructs. In the period of 6 weeks, in histological sections, it was possible to observe that a large part of the biomaterial had already been reabsorbed. However, the critical defect was not completely regenerated, and the neoformed areas remained at the extremities of the defects in all groups. This result is similar to that reported in the study by Auersvald et al. [102], in which critical defects were made in rat calvaria using collagen sponges soaked in 20 µm of teriparatide as a filling biomaterial to evaluate the local performance of the drug after 15 and 60 days. In both periods, the authors reported that the newly formed bone was restricted to the limits of the defect without closure.

In the intragroup histomorphometric analysis over the period of 2 weeks, the groups did not differ statistically (*p* > 0.05). Studies have shown that the local use of teriparatide intermittently and subcutaneously resulted in an increase in the amount of newly formed bone tissue [38,56]. Ozer et al. [42] also addressed the local use of teriparatide with xenografts in rabbits and found that in periods of 4 and 8 weeks, the group with the presence of the drug showed higher mean values of new bone formation. However, these values were greater in the period of 8 weeks. Furthermore, in 4 weeks, the area and that of the defect containing the material with the drug consisted of regions of interconnected bone trabeculae, mainly at the edges of the defect, with loosely collagenized connective tissue, as observed in our study. In the present study, during the 2-week euthanasia period, we suggest that due to the high molecular weight of the teriparatide drug, there was a delay in its total release, which may have impaired the pharmacokinetics [69,103]. Interferences and limitations in the use of bioglass loaded with teriparatide were also reported in the study by Frigério et al. [104] because the incorporation limited the osteoconductive effect of Biogran^®^, and despite the beneficial effects of the drug, the results were less expressive in these groups.

In the intragroup histomorphometric analysis, the neoformed area was greater in the BGTO group in the 6-week period but did not differ statistically from the other groups in the same period (*p* > 0.05). When the intergroup comparison was performed in the 2-week and 6-week periods, it was observed that the BGTO group in the 6-week period statistically differed from the BGO and BGTO groups in the 2-week period (*p* < 0.05), showing an significant increase in bone formation in the area of the defect, thus confirming that the osteoporotic bones of the calvaria are highly metabolically responsive to teriparatide treatment [105]. In the study by Gomes-Ferreira et al. [80], the functionalization with PTH 1–34 in implants did not show a significant effect in healthy animals; however, it played a favorable role when osteoporosis was present. It is possible to suggest that, when evaluating tissue neoformation over longer periods, the total release of the drug and a longer duration of its action in the repaired site can be observed. This slow, intermittent release is interesting, as previous studies have shown that intermittent doses are effective for the anabolic function of the drug [56,106,107].

To investigate the participation of proteins in the bone repair process, immunohistochemistry with osteocalcin (OC) and Rank-L were performed. The immunoexpression of osteocalcin (OC) was more evident in the late period (6 weeks) of bone neoformation, since OC is a non-collagenous protein that participates in the maturation of the mineral part of the bone. This fact has established the study of the maturity of osteoblastic cells, which is a marker of late stages of osteoblastic differentiation [89,108,109]. This result agrees with the study by de Oliveira et al. [39], who presented similar results with more evident immunostaining in late periods of bone tissue repair. The action of the drug on the cellular response in the OVX groups was favorable, since in the period of 6 weeks there was a similarity in the average values of the percentage of immunostained cells for OC in the BGTO group when compared to the CS group, which did not have the interference of osteoporosis. 

Considering that the OVX group has a deficiency in bone repair, there was an increase in the marking of cells related to CO and BGO in a period of 6 weeks, showing that treatment with the drug was able to increase the secretion of this non-collagen protein by osteoblasts, improving osteogenic activity. Although favorable results were observed for the BGTO group, the CS group presented higher or similar mean results when compared to the OVX group, a result that may be comparable to that observed in the study by Luvizuto et al. [58], in which the evaluation of bone repair in the alveoli of ovariectomized rats showed mean values lower than those observed in the Sham group, even in animals treated with raloxifene and estrogen. Another important aspect was the correlation with the results of histomorphometry and histological analysis, which suggested greater bone neoformation in the experimental groups containing the drug in ovariectomized animals, which is confirmed by the increase in CO marking in this group at 6 weeks in comparison. The results of this study also indicated that the cells present in the calvarial bone defects of all groups were positive for RANKL, with the CO group having higher mean values than the CS group. This was expected since osteoporotic bones undergo greater bone resorption when compared to normal bones [40,110,111]. In the 6-week period, it was observed that all groups showed a decrease in the mean values of the percentage of cells marked by RANKL compared to the 2-week period, except for the CS group; this may be related to the fact that the expression of bone markers can be altered by estrogen deficiency and present interference from the evaluated biomaterial [112].

The drug’s ability to stimulate bone formation and increase bone mass and mineral density [45,56,113] corroborate the results of the three-way flexion test points, which demonstrated that maximum force (load/N) in the group of ovariectomized rats with bone defects in tibias filled with bioglass associated with 10% teriparatide (BGTO group) exhibited significantly higher values, statistically differing from the other analyzed groups (*p* < 0.05), which promoted an increase in the biomechanical properties of newly formed bone in rats with endocrine abnormalities when compared to normal animals. Studies with the systemic use of PTH (1–34) for fractures in long bones confirm the increase in mechanical resistance and the amount of newly formed bone [114,115,116,117]. A recent study by Leiblein et al. [118] verified that the maximum force was increased in the biomechanical tests performed in animals treated with PTH (1–34) when compared to animals that received doses of simvastatin, alendronate, and strontium ranelate.

In summary, in our study, the results of the in vitro tests reiterate that the experimental groups were favorable to the differentiation of undifferentiated cells from ovaritomized rats into osteogenic cells. However, we suggest that in the in vivo study, it can be inferred that the calvaria bone has a slower healing process when compared to spongy bone, which is because the calvaria has a smaller blood supply and a smaller amount of medullary bone [119]. We suggest, therefore, that longer evaluation times are necessary to verify the formation of a more mature bone tissue and that, despite this, the biomaterial presented itself as a promising material for use in the bone repair of ovariectomized animals.

## 5. Conclusions

In view of the results obtained and within the experimental conditions of this research, it was concluded that bioglass functionalized with the drug teriparatide 10% prove a positive influence on cell behavior in vitro and bone neoformation in vivo. Additionally, this study contributed to the knowledge of the interactions of osteoblasts with the surfaces of functionalized bio-glasses thus contributing to greater safety in the use of bio-glasses associated with drugs in the health area.

Although these initial results are promising, further studies are needed to evaluate the effectiveness of PTH associated with these biomaterials in new regeneration strategies, to provide additional information on the mechanisms necessary to assess the biological performance of synergistic action between bioglass and the release of the drug teriparatide.

## Figures and Tables

**Figure 1 jfb-14-00093-f001:**
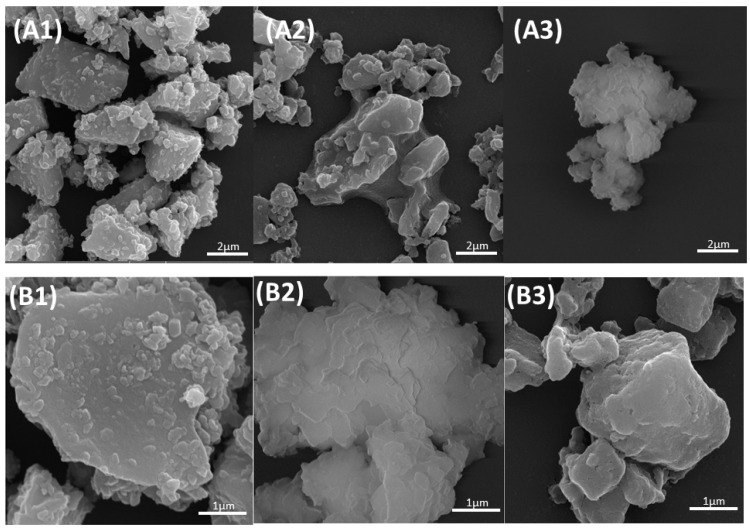
Micrographs of the surfaces of materials. Images obtained by SEM-FEG: (**A1**) 45S5 Bioglass group increase of 20,000 times (scale bars = 2 μm); (**A2**) BG group increase of 20,000 times (scale bars = 2 μm); (**A3**) BGT group increase of 20,000 times (scale bars = 2 μm); (**B1**) 45S5 Bioglass group increase of 50,000 times (scale bars = 1 μm); (**B2**) BG group increase of 50,000 times (scale bars = 1 μm); (**B3**) BGT group increase of 50,000 times (scale bars = 1 μm).

**Figure 2 jfb-14-00093-f002:**
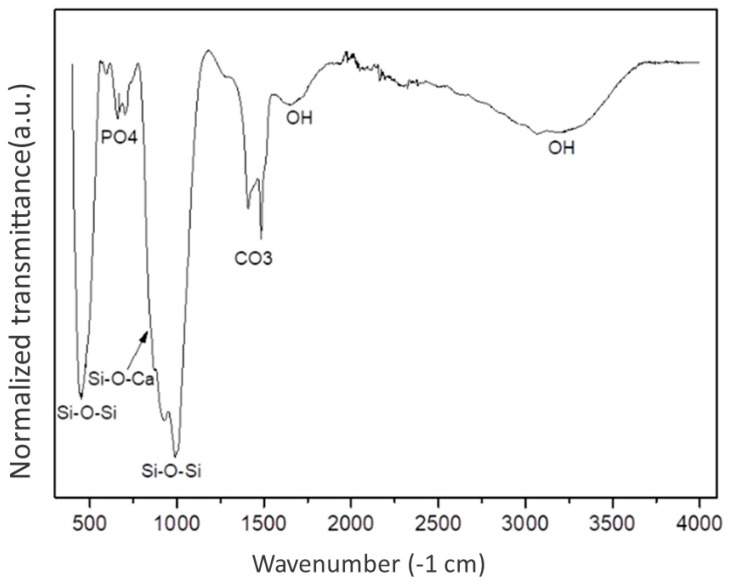
Fourier Transform Infrared Spectrophotometry results.

**Figure 3 jfb-14-00093-f003:**
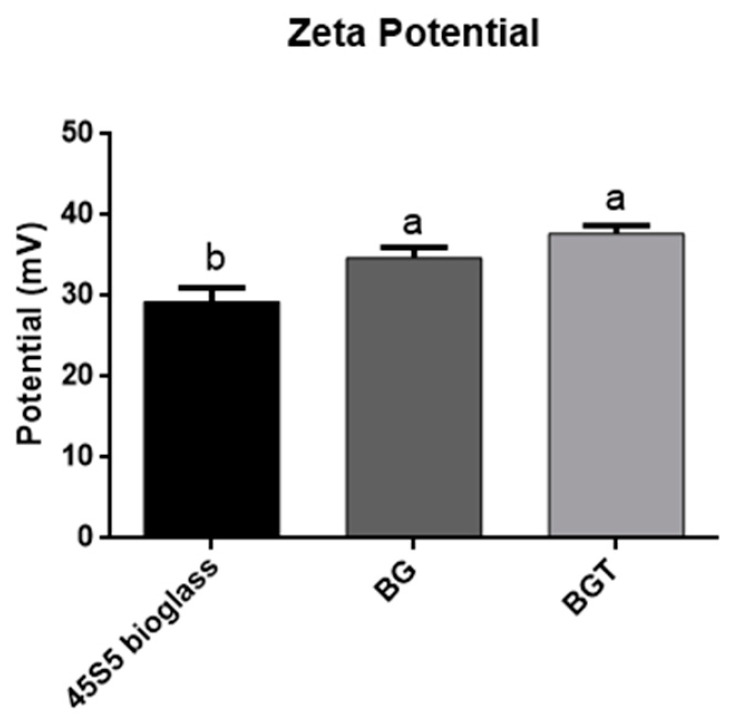
Values of mean and ± standard deviation of the Zeta potential (mV).

**Figure 4 jfb-14-00093-f004:**
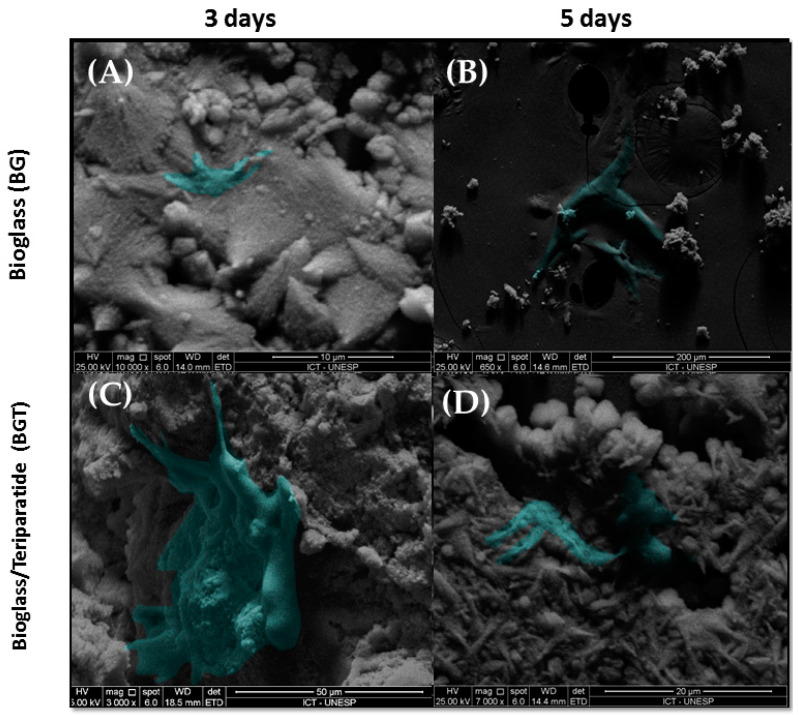
SEM images of cells adherent to the biomaterial: (**A**) BG group in the period of 3 days with an increase of 10,000 times; (scale bars = 10 μm); (**B**) group BG in the period of 5 days with an increase of 650 times (scale bars = 200 μm); (**C**) BGT group in the period of 3 days with an increase of 3000 times (scale bars = 50 μm); (**D**) BGT group within 5 days with an increase of 7000 times (scale bars = 20 μm).

**Figure 5 jfb-14-00093-f005:**
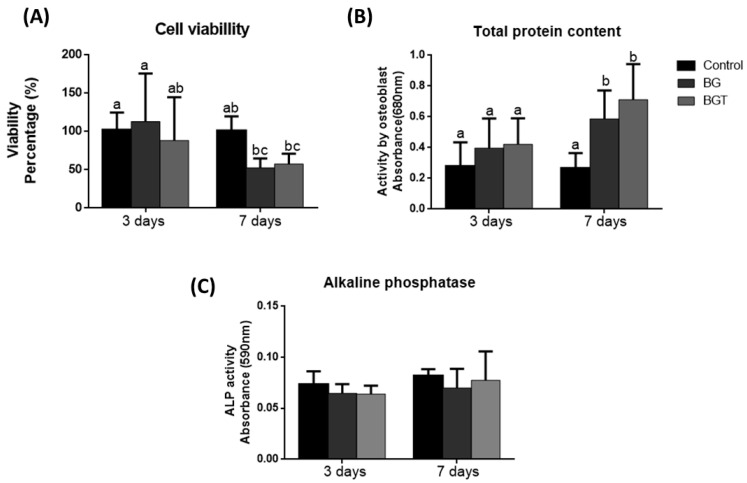
Results obtained through performing in vitro tests. Graphics represent mean values and (±) standard deviation. (**A**) Cell viability (Percentage %) after three days and seven days; (**B**) total protein content (OD—690 nm) after three days and seven days; (**C**) alkaline phosphatase activity (OD—580 nm) after three days and seven days. Letters represent results from Tukey test multiple comparisons; different superscript letters in-dicate significant differences (*p* < 0.05).

**Figure 6 jfb-14-00093-f006:**
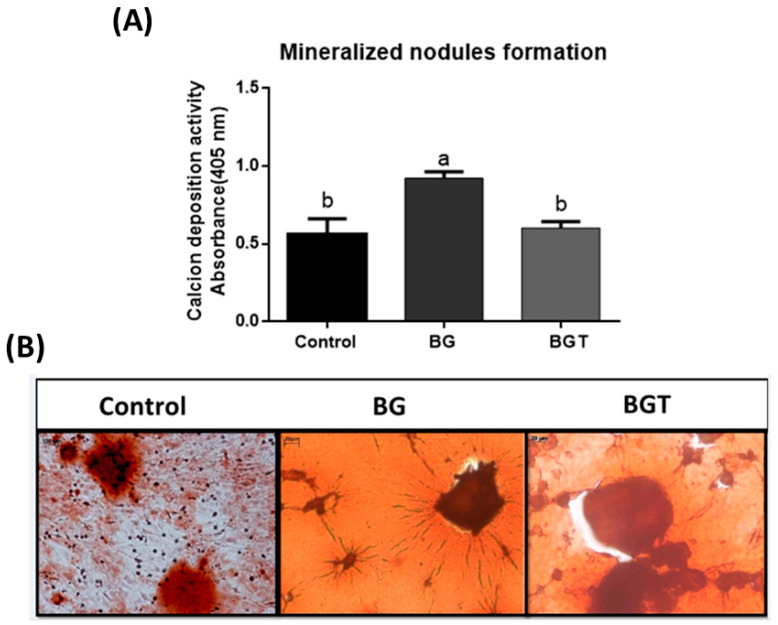
Quantification of calcium in mineralization nodules (OD-405 nm) after 12 days. (**A**) Graphics represent mean values and (±) standard deviation. Letters represent results from Tukey test multiple comparisons; different superscript letters indicate significant differences (*p* < 0.05); (**B**) representation of the formation of mineralization nodules. Cells were stained with red alizarin S to visualize formed calcium deposits. Original magnification ×20 (scale bars = 20 μm).

**Figure 7 jfb-14-00093-f007:**
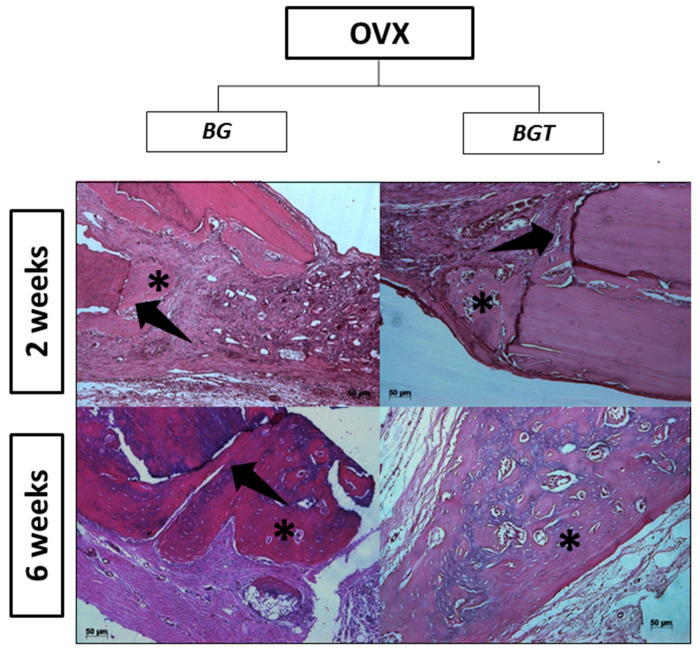
Histological section of the group BGO and BGTO 2 and 6 weeks. Asterisk in the newly formed area. Arrow: black in the delimitation of the critical defect. HE is staining; original magnification ×10 (scale bars = 50 μm).

**Figure 8 jfb-14-00093-f008:**
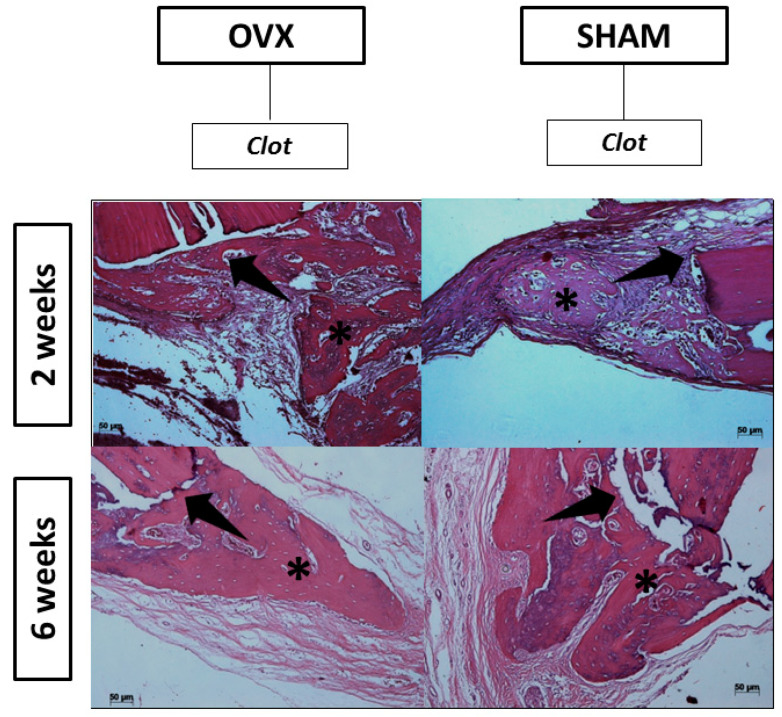
Histological section of the clot group at 2 and 6 weeks. Asterisk in the newly formed area.: Black arrow in the delimitation of the critical defect. HE is staining, original magnification ×10 (scale bars = 50 μm).

**Figure 9 jfb-14-00093-f009:**
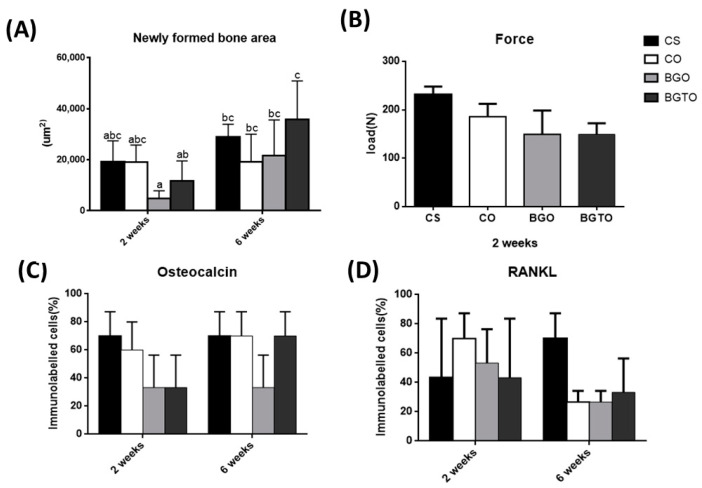
Graphics represent mean values and (±) standard deviation. (**A**) Results obtained by performing bone neoformation in vivo test analyzed at two times: 2 and 6 weeks; Letters represent results from Tukey test multiple comparisons; different superscript letters indicate significant differences (*p* < 0.05). (**B**) force parameter biomechanical test; (**C**) graph showing the distribution of scores (in percentage) referring to the OC immunostaining pattern at 2 weeks and 6 weeks; (**D**) graph showing the distribution of scores (in percentage) referring to the RankL immunostaining pattern at 2 weeks and 6 weeks.

**Figure 10 jfb-14-00093-f010:**
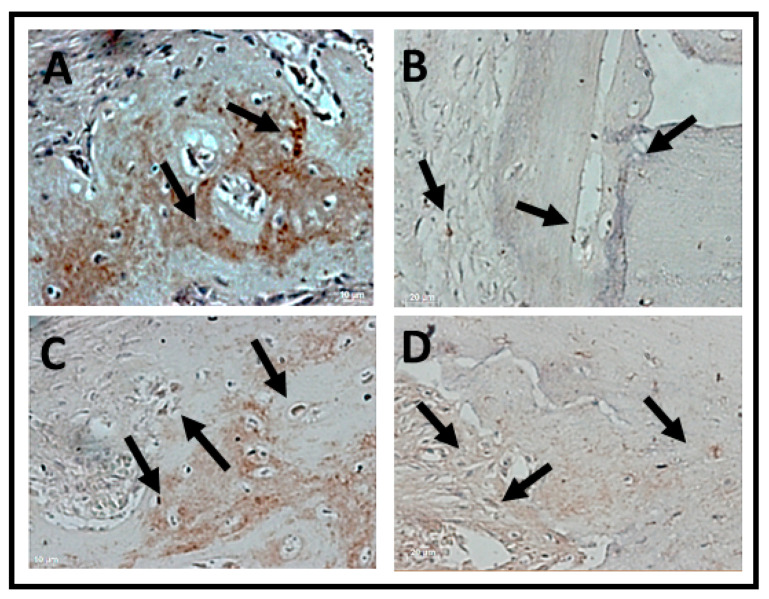
Photomicrographs of the histological sections with positive markings for the polyclonal antibody osteocalcin (OC) over a 2-week period. (**A**) CS, showing evident immunostaiing of cells lining the bone tissue and tissue adjacent to the bone; (**B**) CO, positive immunostaining of OC compatible with the beginning of the bone tissue healing process; (**C**) BGO, immunostaining of OC in adjacent tissue and few lining cells in bone tissue; (**D**) BGTO, osteocytes and lining cells in adjacent tissue are marked by OC, showing few marked regions compatible with bone immaturity in this period. Osteocytes were positive for OC labeling in all groups. Arrows indicate areas and label cells immunomarked by osteocalcin. Original magnification ×40 (scale bars = 20 μm.)

**Figure 11 jfb-14-00093-f011:**
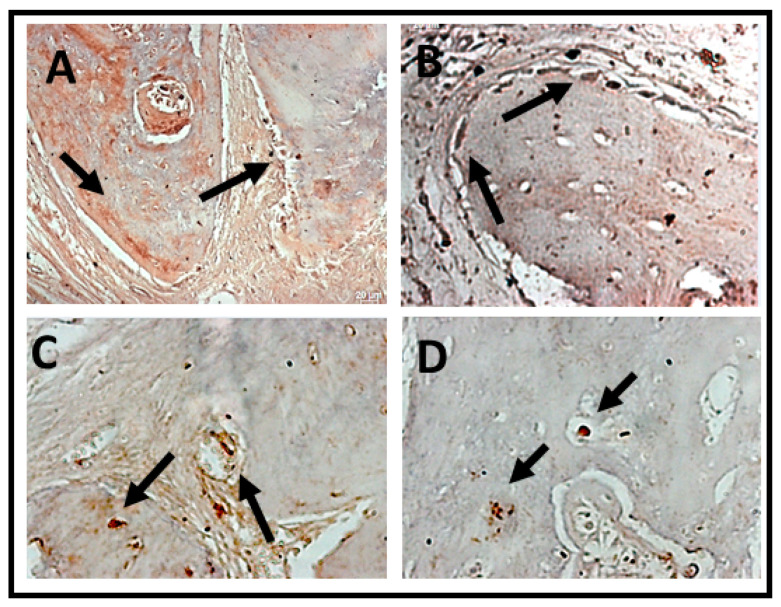
Photomicrographs of the histological sections with positive markings for the polyclonal antibody osteocalcin (OC) over a 6-week period. (**A**) CS, immunostaining osteocalcin with mineralized bone matrix; (**B**) CO, osteoblastic cells around the newly formed bone tissue; (**C**) BGO, immunostaining of OC in cells in bone tissue; (**D**) BGTO, osteocytes within the gaps in the newly formed bone tissue. Osteocytes and osteoblast were positive for OC labeling in all groups. Arrows indicate areas and label cells immunomarked by osteocalcin. Original magnification ×40 (scale bars = 20 μm.)

**Figure 12 jfb-14-00093-f012:**
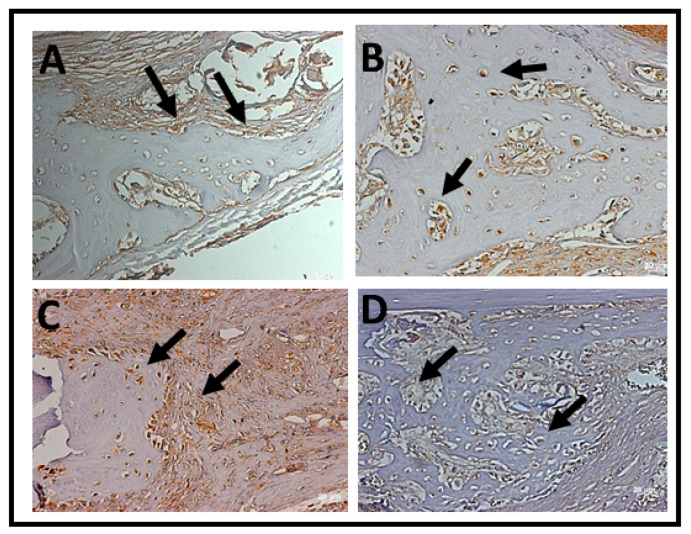
Photomicrographs of the histological sections with positive markings for the polyclonal antibody RankL over a 2-week period. (**A**) CS, immunostaining in the nucleus of osteoblasts in the periphery of the neoformed bone and in fibroblasts present in the connective tissue; (**B**) CO, immunostaining in the nucleus of newly formed osteocytes in the neoformed bone matrix; (**C**) BGO, evident markings in connective tissue and in osteocytes and osteoblasts present in the matrix and in the periphery of the bone neoformed; (**D**) BGTO, markings on osteocytes in the newly formed matrix and less evident in the adjacent connective tissue. Arrows indicate areas and label cells immunomarked by RankL. Original magnification ×40 (scale bars = 20 μm.)

**Figure 13 jfb-14-00093-f013:**
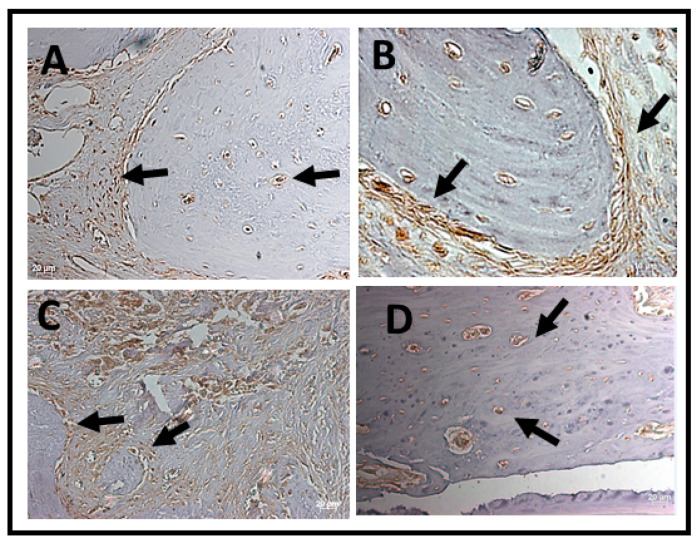
Photomicrographs of the histological sections with positive markings for the polyclonal antibody RankL over a 6-week period. (**A**) CS, more evident markings on connective tissue and little marking in osteocytes and osteoblasts present in the matrix and in the periphery of the newly formed bone; (**B**) CO, positive marking on connective tissue and on cells from the periphery of the neoformed bone; (**C**) BGO, connective tissue with evident RankL immunostained areas; (**D**) BGTO, RankL immunostaining that is less evident in connective tissue and few osteocytes with evident marking. Arrows indicate areas and label cells immunomarked by RankL. Original magnification ×40 (scale bars = 20 μm).

**Table 1 jfb-14-00093-t001:** Energy-dispersive analysis (EDS).

EDS Groups	O (Wt%)	Na (Wt%)	Ca (Wt%)	Si (Wt%)	Au (Wt%)	P (Wt%)	C (Wt%)
45S5 bioglass	47.8 ± 0.2	11.7 ± 0.1	12.7 ± 0.1	11.3 ± 0.1	0	1.6 ± 0	0
BG	45.2 ± 0.2	10.5 ± 0.1	13.9 ± 0.1	9.8 ± 0.1	2.5 ± 0.2	1.4 ± 0	16.6 ± 0.2
BGT	49.8 ± 0.2	24.4 ± 0.2	9.7 ± 0.1	6.5 ± 0.1	6.2 ± 0.3	3.4 ± 0.1	0

Mean and standard deviation values of measures’ percentage of mass (Wt%).

**Table 2 jfb-14-00093-t002:** Scores of immunohistochemical Analysis established for experimental groups (OC and RankL) at 2 and 6 weeks, showing: (+) = positive; (c) (++) = superpositive.

	OC		RankL	
	2 Weeks	6 Weeks	2 Weeks	6 Weeks
CS	(++)	(++)	(+)	(++)
CO	(++)	(++)	(++)	(+)
BGO	(+)	(+)	(+)	(+)
BGTO	(+)	(++)	(++)	(+)

## Data Availability

All raw data from characterization are available from the corresponding author upon request.

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
