# Peer review of "The Local Release of Teriparatide Incorporated in 45S5 Bioglass Promotes a Beneficial Effect on Osteogenic Cells and Bone Repair in Calvarial Defects in Ovariectomized Rats"

_jfb, 2023, doi:10.3390/jfb14020093_

Round 1

Reviewer 1 Report

The manuscript (The local release of teriparatide incorporated in 45S5 bioglass promotes a beneficial effect on osteogenic cells and bone repair in calvarial defects in ovariectomized rats) is intesting.

However, some minor revisions have to be considered before recommending its acceptance for publication.

1- Some compelx sentences arise in this article. Simpler, more comprehensive ones will be better.

The scales of Fig. 1 can not be recognized. Clearer ones have to be listed.

2- Fig.3. It is (Zeta petentail) not potentail zeta. Please correct. 

3- The recent relevant references are not enough. Relevant refs. published in 2022 shoul be cited here.

Reviewer 2 Report

In this study, 45S5 bioglass was manufactured by melting-quenching functionalized bioglass (BG) and functionalized bioglass with 10% teriparatide (BGT) using the sonochemical technique. In vitro tests were conducted on osteoblasts derived from mesenchymal cells isolated from the femurs of ovariectomized rats. The study's findings indicate that bioglass functionalized with 10% teriparatide has a positive effect on cell behavior in vitro and bone neoformation in vivo. In addition, this study contributed to our understanding of the interactions between osteoblasts and the surfaces of functionalized bio-glasses, thereby enhancing the safety of bio-glasses used in conjunction with pharmaceuticals. Furthermore, the abstract of the manuscript, introduction, and conclusions are good, and the results are adequately discussed. But I have some criticisms about the article below. After these corrections are made by the Authors, the manuscript may be accepted for publishing in the Journal of Functional Biomaterials.

My comments to authors:

1. There are a few grammatical errors in the English language in the manuscript. Please check and correct.

2. You produced the bioglasses by the melt quenching method and then you ground these produced bioglasses to 200 mesh. Then, you applied drugs to these ground micron-sized powders and did your biological tests. I wonder why did you use ground bioglass instead of glass in the form of a piece in experiments? Could you please explain it?

4. Some recent papers about soda-lime silicate glass as bioglass should be added to the introduction part /or Materials and Methods of the manuscript. For examples:

DOI: 10.12693/APhysPolA.132.442

Reviewer 3 Report

The study reports the fabrication of teriparatide-coated bioglasses for overcoming calvarial defects in ovariectomized rats. The authors provide compelling experimental evidence that demonstrates bioglass-functionalized teriparatide influences cell behavior in vivo. This report also suggests a greater safety in the use of bio-glasses associated with drugs in treating osteoporosis. I think this study can be published in the journal after addressing the following minor concerns by this reviewer.

1.     Page 2, line 83: change “osseonecrotic” to “osteonecrotic”

2.     Figure 2, change the x and y-axis titles to Wavenumber (cm-1) and Normalized transmittance (a.u.), respectively.

3.     In the discussion about zeta potential measurements, the authors mention the term ‘modulus value’. Though the term is scientifically correct, this reviewer suggests replacing ‘modulus value’ with ‘magnitude’. This change must be done in other places where the term modulus is used.

4.     Add the y-axis title and its unit to Figure 3.

5.     In Figure 4, it is challenging to see the cells for novice readers. Can the authors mark the periphery of cells in the SEM images for better visualization?

6.     Page 12, lines 435 – 441: This paragraph is confusing to the reader. If there is no statistical significance in the p-value range authors use, authors should not claim higher or lower cell viability for one sample group.

7.     Why is the total protein content higher for BG and BGT compared to the control? The authors should provide an explanation in the discussion section.

8.     Figures 5 and 6 require an appropriate y-axis title. The graphs must be able to provide stand-alone complete information even in the absence of figure captions. It is not acceptable to write percentage or wavelength as y-axis titles.

9.     Why is calcium mineralization higher for BG than other groups? The authors should provide an explanation in the discussion section.
